# Identification of *Clec4b* as a novel regulator of bystander activation of auto-reactive T cells and autoimmune disease

**Liselotte Bäckdahl** [1©], **Mike Aoun**[1©], **Ulrika Norin**[1], **Rikard Holmdahl**[1,2]*

**1** Medical Inflammation Research, Department of Medical Biochemistry and Biophysics, Karolinska Institute, Stockholm, Sweden, **2** The Second Affiliated Hospital of Xi'an Jiaotong University (Xibei Hospital), Xi'an, China

© These authors contributed equally to this work.

* rikard.holmdahl@ki.se

⬛ OPEN ACCESS

**Data Availability Statement:** All relevant data are within the manuscript and its Supporting Information files.

## Abstract

The control of chronic inflammation is dependent on the possibility of limiting bystander activation of autoreactive and potentially pathogenic T cells. We have identified a non-sense loss of function single nucleotide polymorphism in the C-type lectin receptor, Clec4b, and have shown that it controls chronic autoimmune arthritis in rat models of rheumatoid arthritis. *Clec4b* is specifically expressed in CD4$^+$ myeloid cells, mainly classical dendritic cells (DCs), and is defined by the markers CD4$^+$/MHCII$^{hi}$/CD11b/c$^+$. We found that *Clec4b* limited the activation of arthritogenic CD4+αβT cells and the absence of *Clec4b* allowed development of arthritis already 5 days after adjuvant injection. *Clec4b* sufficient CD4$^+$ myeloid dendritic cells successfully limited the arthritogenic T cell expansion immediately after activation both *in vitro* and *in vivo*. We conclude that *Clec4b* expressed on CD4+ myeloid dendritic cells regulate the expansion of auto-reactive and potentially pathogenic T cells during an immune response, demonstrating an early checkpoint control mechanism to avoid autoimmunity leading to chronic inflammation.

## Author summary

To identify early disease regulatory mechanisms in autoimmune diseases such as rheumatoid arthritis (RA) is challenging not only because of the genetic and environmental complexity but also because of the critical autoimmune time-period that precedes the clinical diagnosis. Therefore, we set out to study the complex disease pathways in a more restricted setting. Through genetic segregation of rat crosses, followed by the selection of recombinants to produce minimal congenic strains, we have identified a single nucleotide polymorphism regulating the expression of Clec4b2 that in turn controls the development of arthritis. The Clec4b gene is normally expressed in a population of antigen-presenting cells that can limit enhanced activation of bystander autoreactive T cells during an immune-priming response. This previously unknown type of immune regulation reveals the existence of a mechanism protecting against autoimmune dieases by the avoidance of

**Funding:** This work was supported by grants from the Knut and Alice Wallenberg Foundation, the Swedish Association against Rheumatism, the Swedish Research Council, and the Swedish Foundation for Strategic Research. The research leading to these results has further received funding from the European Community's IMI project BTCURE. The funders had no role in study design, data collection and analysis, decision to publish, or preparation of the manuscript.

**Competing interests:** The authors have declared that no competing interests exist.

bystander activation of autoreactive T cells during a normal immune response to foreign antigen.

## Introduction

A tissue-specific autoimmune disease process starts decades before the clinical onset of auto-immune diseases, such as rheumatoid arthritis (RA) [1]. Most likely the first trigger involves the activation of autoreactive T cells, which are normally regulatory or anergic, into a more aggressive state. The activation requires strong costiumulation, which during an immune response is mediated by adjuvants carried by infectious organisms or possibly damaged endogenous cells, or environmental hazards such as tobacco smoke [2]. These challenges trigger the innate immune system, leading to the activation of autoreactive T cells. Innate immune cells interpret infectious intruders or danger signals *via* a range of pattern-recognizing receptors (PRRs) on their cell surfaces. When the innate cells sense enhanced risk in the environment, these cells are able to activate other cells, such as T cells. If the activation of adaptive responses displays joint specificity, the situation could initiate clinical arthritis. Animal models of arthritis mimic these disease stages [3]. They are initiated by adjuvant immunization followed by an autoimmune response to a tissue-specific protein. In the case of collagen-induced arthritis (CIA), it is the type II collagen (CII) that is involved and in the case of arthritis induced by various type of adjuvants, such as pristane-induced arthritis (PIA), or mineral-oil induced arthritis (OIA), a bystander response is raised to a pattern of unknown endogenous auto-antigens [4]. Clinical arthritis starts to develop 2 weeks after the injection, as a result of an inflammatory attack on peripheral cartilaginous joints, involving the autoimmune response, that later can develops into a chronic inflammatory disease.

To determine the basic mechanisms leading to an autoimmune disease we searched for the genetic polymorphisms that allow the development of disease in certain inbred strains. For our investigation, we selected a cross between the DA rat, which is highly susceptible to auto-immune diseases, and the disease resistant E3 rat strain. The rats were injected intra-dermally with pristane, a simple alkene adjuvant oil which triggers a disease that fulfils the classification criteria for RA [5]. Through genetic linkage mapping, we identified 20 arthritis-associated loci in the DA rat [6]. One of the major loci was localized to chromosome 4 and was denoted *Pia7*. The same locus was identified using several different arthritis models, including CIA, PIA and OIA [7–9]. With the use of recombinant congenic strains, the disease association could be mapped to a gene family encoding C-type lectin receptors, denoted hereafter as APLEC (anti-gen presenting cell expressing lectin like receptor gene complex) [10,11]. APLEC contains a cluster of 8 genes, 4 of which have a carbohydrate recognition domain highly similar to the full-length *Clec4a/Dcir and* are thus denoted *Clec4a1-3*. The complex also covers 4 non-*Dcir* like C-type lectins: *Clec4b2b/Dcar*, *Clec4d/MCL*, *Clec4e/Mincle* and *pClecn/Dectin2* (a non-functional pseudo-gene in both the DA and E3 strains). It has been proposed that both *Clec4a* and *Clec4b* are important in antigen presentation and uptake [12,13] as well as in T cell responses [14] and can be assumed to be of importance early in an immune response. Furthermore, there is a genetic association to human DCIR in RA [15]. In contrast, *Clec4d* and *Clec4e* are involved in later inflammatory responses [16]. We have now located a single nucleotide polymorphism (SNP) in the *Pia7* locus, regulating the *Clec4b* gene. Only E3 derived cells express the Clec4b/Dcar receptor and it is expressed on CD4$^+$ Dendritic cells. The loss of *Clec4b/Dcar1* expression leads to dysregulation of CD4$^+$ myeloid antigen presenting cells, which lose the ability to control bystander activation of autoreactive T cells. This previously

unknown physiological regulator of autoreactive T cells, are of critical importance to the development of autoimmune disease.

## Results

### A SNP within the *Clec4b* gene explain the arthritis-association

The DA.E3-*Pia7* congenic strain is less susceptible to arthritis compared to the DA strain [11]. We chose to fine map the underlying polymorphisms within the Pia7 locus for association with arthritis using OIA, which gives a milder arthritis compared with PIA. In fact, the same locus as Pia7 has also been mapped in OIA and designated Oia2 [7]. We aimed to produce recombinant congenic strains to identify the experimental arthritis regulating polymorphism in Pia7. Whole-genome sequencing of both the DA and the E3 genomes was advantageous as it enabled us to identify at least 1 microsatellite marker/50 Kb within the APLEC region [6]. More than 5000 pups were screened for recombination events using a panel of 37 microsatellite markers in the original 1.5 Mb DA.E3-*Pia7* interval. Two recombinant congenic strains were produced. The first identified sub-congenic strains, DA.E3-*Clec4a* contained arthritis protective E3 alleles only in the 4 *Clec4a* isoforms of the APLEC region where the recombination event occurred immediately after the Clec4a gene but before Clec4b2 between MS*Aplec*710 and MS*Aplec*740. The DA.E3-*Clec4bde* sub-congenic strain was the product of two separate recombination events, containing E3 alleles in *Clec4a*, *Clec4b2*, *Clec4d*, *Clec4e*, and the *Clec4n/Dectin2*, which is a pseudogene in both DA and E3 (Fig 1A). Limited on the centromeric side by a recombination between MS*Aplec*573 and MS*Aplec*633 and distally by a recombination event occurring right after *Clec4e* between the markers D4rat90 and MS*Aplec*981.

OIA assessment in the DA.E3-*Clec4a* strain showed no protection within this region, contrary to the full DA.E3-*Pia7* congenic strain (Fig 1C). The sub-congenic strain DA.E3-*Clec4bde* on the other hand showed almost complete protection against OIA, as well as to PIA and CIA (Fig 1C–1E). Sequencing the DA.E3-*Clec4bde* fragment identified two coding polymorphisms: a nonsense mutation in the DA strain that causes a truncated protein at exon two of *Clec4b2*; and a non-synonymous SNP in the membrane domain of *Clec4e* (Fig 1B). The mouse has two isoforms of *Clec4b*; *Clec4b1* and *Clec4b2*. Since the rat only has the *Clec4b2* homolog we refer to this gene as *Clec4b*. The *Clec4d* gene was excluded as a candidate since it does not contain any coding or regulatory polymorphisms and does not show any differences in gene expression between strains (Fig 1F). A comparison of the genome sequence with other inbred rat strains showed that the arthritis protective PVG and E3 share the same *Clec4bde* haplotype [6], making them distinct from the DA *Clec4bde* haplotype. The arthritis protective F344 strain, on the other hand, have a *Clec4bde* haplotype that is identical to DA. When trying to identify strains that deviate in their *Clec4bde* haplotype, only 1 strain was different. The BN strain shares alleles with DA only in the *Clec4e* locus (see Fig 1B), whereas the remaining sequence of the BN *Clec4bde* region is shared with E3, the arthritis protective haplotype. In a literature search to compare F2 crosses between DA and PVG, and between DA and BN strains, the arthritis quantitative trait locus at APLEC was reproduced in both crosses despite the arthritis susceptible version of *Clec4e* in BN. An APLEC arthritis QTL could not be reproduced in crosses between DA and the arthritis protective F344 strain, principally because F344 has the DA version of the full APLEC [9]. The inherent ability to reproduce linkage in an F2 cross between DA and BN provided sufficient evidence to us to conclude that arthritis regulation most probably stems from variations in *Clec4b* rather than *Clec4e*. We identified a causative nonsense SNP in the *Clec4b* DA allele that causes a truncation of the DA allelic protein that is nonfunctional but with expressed transcripts. Thus, the DA strain is a natural functional knock-out for Clec4b, which further substantiates the gene as the arthritis regulating gene in

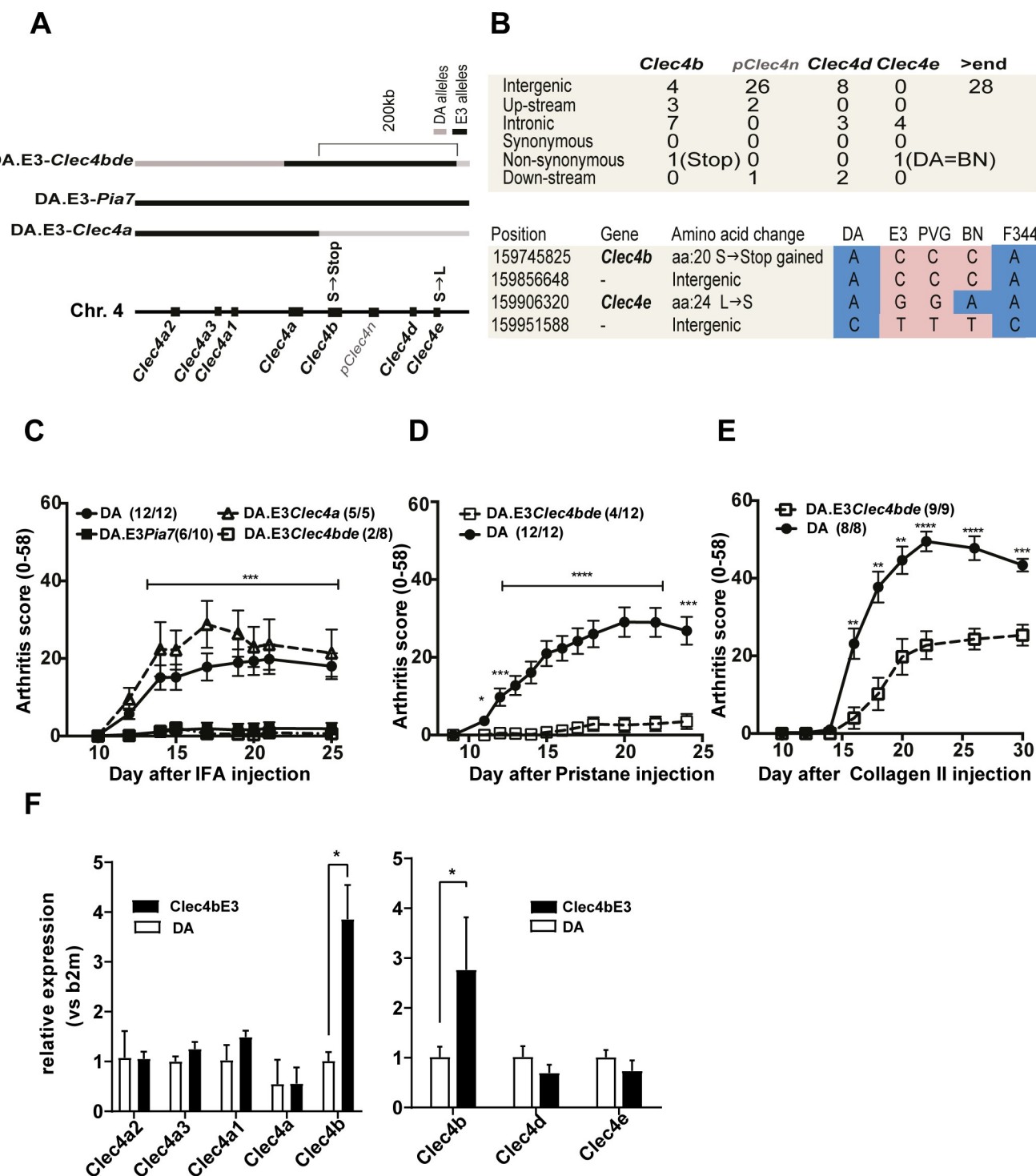

**Fig 1. Identification of the arthritis-regulating gene in the *Oia2/Pia7/Cia13* locus on rat chromosome 4.** a) An illustration of the distribution of the 8 C-type lectin genes across the APLEC interval on rat chromosome 4 and the congenic fragments with genomic regions harboring arthritis-susceptible DA alleles (grey) and arthritis-protective E3 alleles (black) for the three congenic strains DA.E3-*Pia7* to the far right, DA.E3-*Clec4a* to the left of the chromosome and DA.E3-*Clec4bde* to the far left. b) An illustration of the distribution of SNPs across the DA.E3-*Clec4bde* congenic fragment[6] and the *Clec4bde* encoded genes, together with an illustration of the two non-synonymous SNPs and nearby SNPs in five different inbred strains including the reference rat strain BN. c) OIA development in DA strain and the 3 congenic strains. d) PIA development over time in the original DA strain and DA.E3-*Clec4bde* congenic. e) CIA development over time for the DA strain and the DA.E3-*Clec4bde* strain. Significance stars illustrate Mann. Whitney Test * = $p<0.05$, ** = $p<0.01$, *** = $p<0.001$ **** = $p<0.0001$ and SEM. f) Gene expression of all APLEC encoded genes for DA and *Clec4b*[E3] in naïve splenocytes. House-keeping gene is b2m and all genes are compared to one DA sample.

the APLEC region. The two DA strains with different *Clec4b* alleles are hereafter denoted DA and DA.*Clec4b*^E3, abbreviated as *Clec4b*^E3 and occasionally *E3. To investigate if other non-coding variations from the fragment could be regulatory, expression of all APLEC encoded genes were measured in DA and *Clec4b*^E3 in naïve splenocytes. Only Clec4b showed differential expression (Fig 1F). All raw data to figures are available in the S1 Data file.

## *Clec4b* is expressed in steady-state/immature CD4⁺ myeloid-derived spleen cells

In order to identify possible mechanisms by which *Clec4b* could regulate experimental arthritis we first aimed to identify which cell types express the *Clec4b* gene. We analysed gene expression in blood, lymph nodes, thymus, bone marrow and spleen (Fig 2A). The highest *Clec4b* expression was found in the spleen, but lower expression levels could also be observed in blood and lymph nodes. Bone marrow showed lower expression compared to spleen whereas

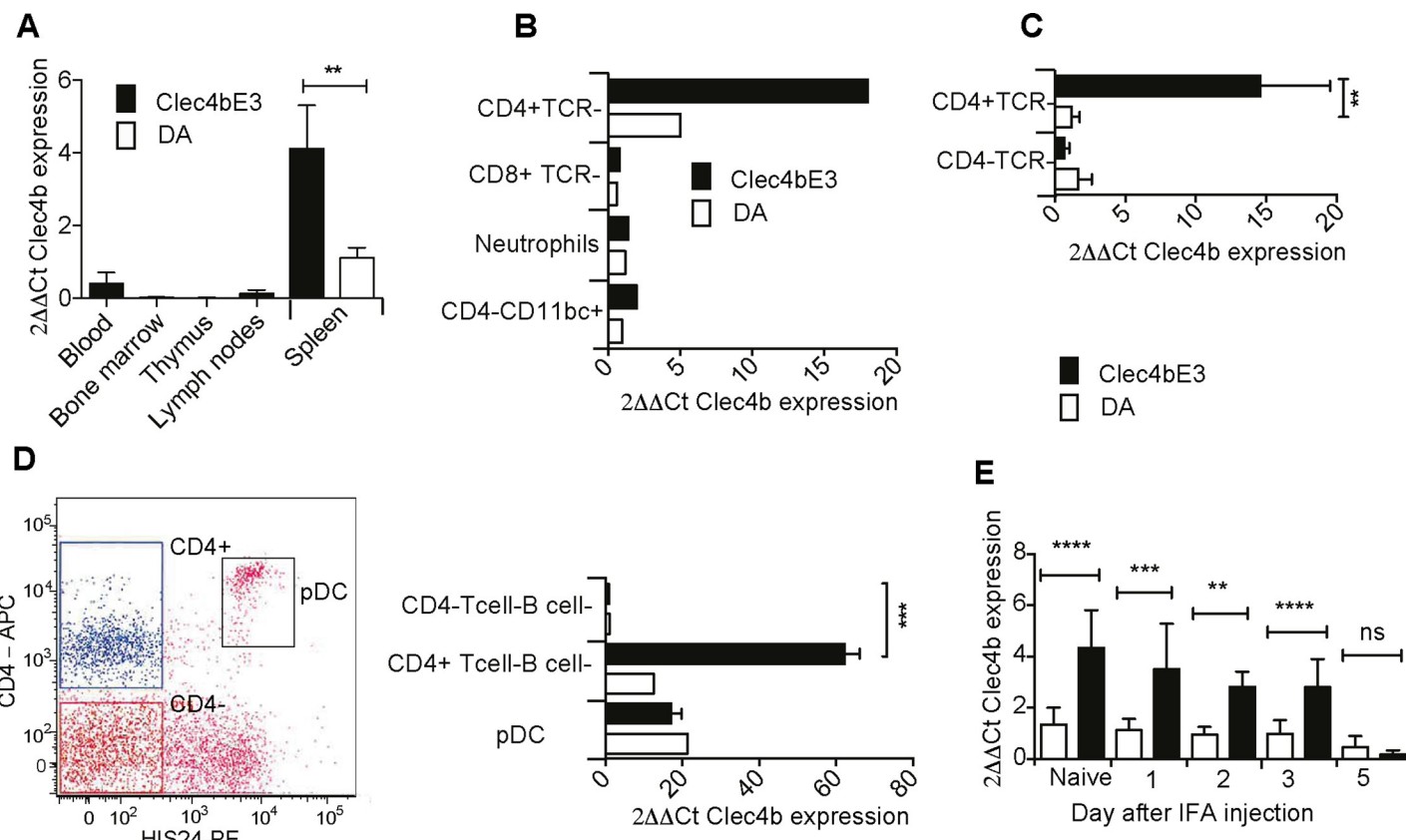

**Fig 2. *Clec4b* expression is restricted to immature CD4+ myeloid cells in the spleen.** a) Relative expression of *Clec4b* in five naïve immune-related tissue; spleen, blood, thymus, bone-marrow and inguinal lymph nodes (*n = 5* for spleen and thymus, remaining tissues *n = 3*). b) *Clec4b* expression of naïve spleen cells. The samples were pooled from three different spleens. Cells were first negatively selected for TCR, and then positively selected for CD4⁺ and CD8⁺ cells, respectively. After CD4/CD8 sorting, the remaining cells were selected for CD11b/c. Neutrophils were enriched from blood using Ficoll-Hypaque centrifugation. c) Non-pooled spleen samples were negatively selected for T cells and positively for CD4. The CD4- fraction is the flow-through cell (*n = 5*). d) Relative *Clec4b* expression in three naïve *Clec4b*^E3 or one DA sample, all sorted with a Mo-flow FACS sorter. The FACS plot depicts the gating strategy for the three assessed population. Before CD4 and His24 (CD45R), the population determining gating, the cells were negatively sorted by gating for TCR and CD45RA, eliminating T cells and B cells. The cells were His24⁻ CD4⁺ (CD4+ DC), His24⁻ CD4⁻ (CD4⁻ DC) or His24⁺ CD4⁺ (pDCs) (*n = 3* for *Clec4b*^E3 and *n = 1* for DA). e) Relative *Clec4b* expression in naïve and oil-primed splenocytes selected for TCR-CD4+ and for day 0, day 1 day 2, day 3, and day 5 after oil injection. *n = 5* for all groups. Significance stars depict Mann. Whitney Test * = *p<0.05*, ** = *p<0.01*, *** = *p<0.001* and SEM.

no expression could be detected in thymus. *Clec4b* gene-expression was significantly down regulated in *DA* spleen compared to the *Clec4b*[E3] strain.

Next, we aimed to identify which splenic cell population expresses *Clec4b*. First, we pooled three naïve *Clec4b*[E3] and three DA spleens and selected T cells and B cells using anti-pan-T and B cell labelled magnetic beads. The T and B cell negative flow-through was elected in two new magnetic labelled fractions, one for CD4$^+$ and another for CD8$^+$. The CD4$^-$ flow-through was lastly also selected for CD11b/c$^+$ cells. *Clec4b* expression was low in all cell types tested except for CD4$^+$ T and B cell negative cells (Fig 2B). Then unpooled individual magnetic bead selected splenocyte samples were assessed, *Clec4b* expression was significantly enriched in cells from *Clec4b*[E3] compared to DA, and more than a tenfold increase in CD4$^+$TCR$^-$ compared to the CD4$^-$TCR$^-$ cells was observed (Fig 2C). More than a sixtyfold difference in *Clec4b* expression was confirmed between CD4$^+$ and CD4$^-$ non-T/B cells when sorted with a flow sorter. This showed *Clec4b* expression exclusively in CD4+αβTCR$^-$CD45RA$^-$ (a B cell-marker) cells, whereas CD4$^-$TCR$^-$CD45RA$^-$ had close to undetectable levels of *Clec4b* (Fig 2D). Interestingly, plasmacytoid dendritic cells (pDCs) are known to be CD4$^+$ in the rat, and to determine if pDCs also express *Clec4b* we used a previously established rat pDC sorting protocol [17] followed by gene-expression assay. Lower levels of *Clec4b* were observed in pDCs, albeit without any strain difference (Fig 2D). A recent study by Daws et al [18] supports both of these findings, suggesting that rat Clec4b is highly expressed in CD4$^+$ myeloid cells but not in CD4$^+$ pDCs.

The fact that OIA is a rapid arthritis disease model, with an onset of clinical arthritis within 10–12 days of injection, suggests that disease regulating genes are already operating within the first few days of injection. In order to examine changes in the kinetic expression of *Clec4b* during early arthritis-induction, CD4$^+$ TCR$^-$ bead selected splenocytes were isolated from both naïve animals and at different time points after mineral oil injection. The highest levels of *Clec4b* expression were observed in naïve splenocytes. A daily decrease in gene-expression was shown until day 5 after injection, when *Clec4b* levels were close to undetectable (Fig 2E). This supports the argument that regulation of *Clec4b* expression is sensitive to rapid environmental changes, either transmitted by instant general immune responses to the oil injection. Alternatively, it indicates that the receptor is sensitive to the oil itself. The rapid downregulation of *Clec4b* after oil injection suggests that the C-type lectin receptor is expressed in naïve or steady-state DCs as a possible steady-state off-switch and that this signal could be important for accurate immune-activation.

## The CD4$^+$*Clec4b*$^+$ cell population consists of two distinct subtypes, classical and granular/plasmacytoid dendritic cells

The *Clec4b* gene is expressed in CD4$^+$ cells that are TCR$^-$. CD4 is frequently expressed on pDCs in both rats and humans, [17,19]. It has been suggested that CD4 is expressed on a subpopulation of conventional dendritic cells, cDC2 [20] but this population is poorly characterized in the rat. Rat CD4$^+$ DCs have been described as being better regulators of the MHCII$^+$ Th1 adaptive responses than CD4$^-$ DCs [21], therefore, we wanted to investigate the phenotype and function of the CD4$^+$TCR$^-$ DC population. The CD4$^+$*Clec4b*$^+$ population was selected from naïve splenocytes with magnetic labelled pan T cell microbeads and, in the second step, with beads isolated TCR$^-$CD4$^+$ cells. The selected cells were then labelled with fluorescent antibodies and studied in a flow cytometer.

The yield from 100 x10$^6$ splenocytes was 3.0 x 10$^6$ ± 0.3 CD4$^+$TCR$^-$ cells and 26.1x 10$^6$ ± 3.6 CD4$^-$TCR$^-$. More than 80% of the CD4$^+$ selected fraction expressed CD4 and had less than 2% T cells (Fig 3A and 3B). Within the CD4$^+$ selected cell fraction, CD11b/c and the granulocyte

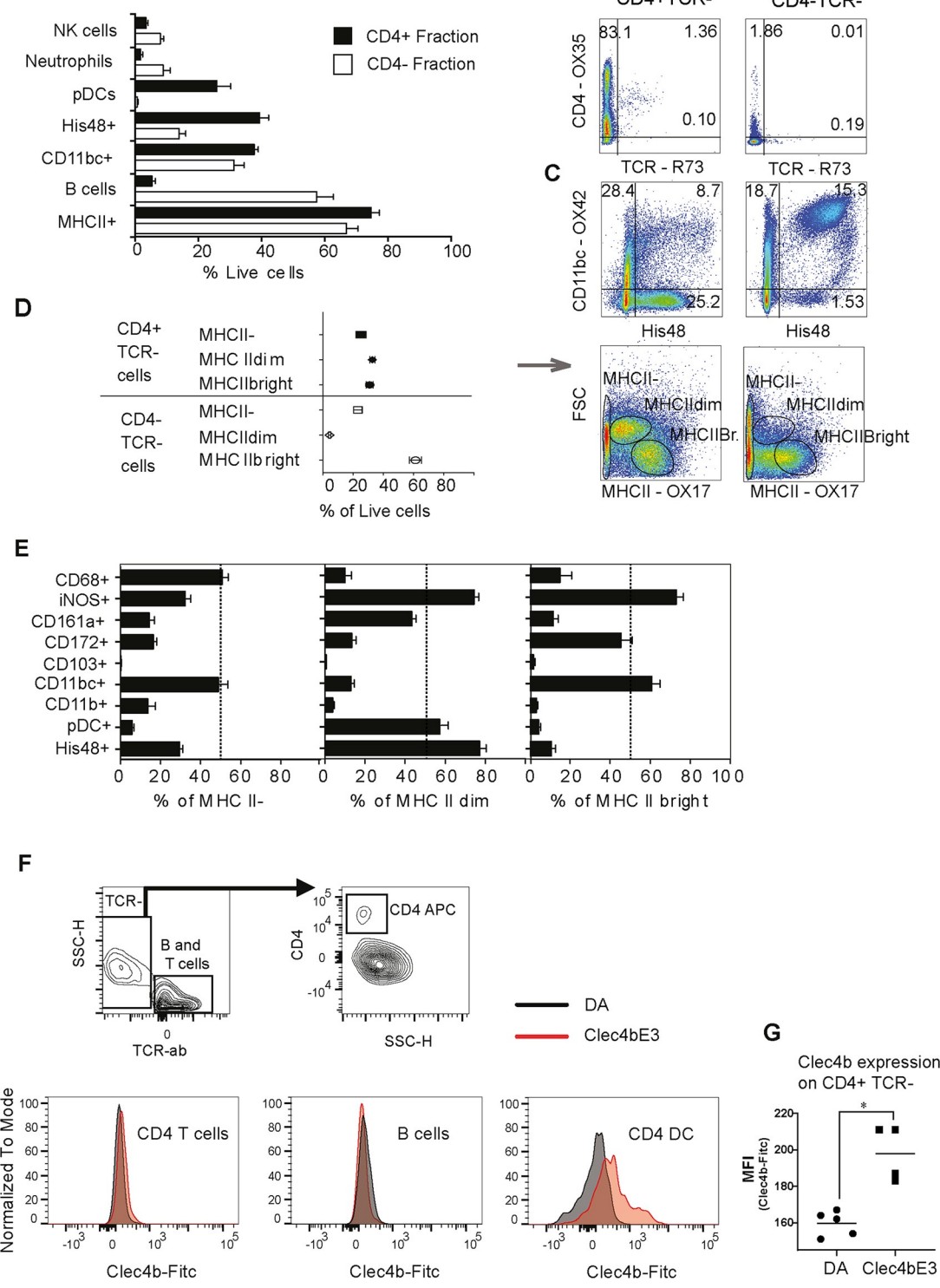

**Fig 3. _Clec4b_ is expressed in CD4+ MHCII+ DCs in two distinct sub-populations that are either MHCIIhi classical dendritic cells or MHCIIdim pDCs.** a) Frequencies of florescence labeled cells (n = 8 for all groups), first selected to be TCR-, and after a second sorting separated into either CD4[+] or CD4[-] fractions using a CD4 antibody coated beads. NK cells are CD161hi, Neutrophils are His48+ and CD11b/c+, pDC are CD4+ and (CD45R) His24[+], B cells are CD45RA[+] (OX33), CD11b/c cells are OX42[+] and His48 cells are granulocytes. b-d) Antibody labeled cell frequencies for CD4[+] and CD4[-] fractions in the total mononuclear cell population. b) Shows CD4 and TCR (clone R73) expression. c) Expression of CD11b/c and His48. d) Illustrate MHCII[+] expression (clone OX17) separated by cellular size by the forward scatter, and the three populations identified are also

analyzed for each of the 8 samples and the cell frequencies are illustrated in a box-plot graph. e) Illustration of the cellular distribution based on antibody labeled fluorescence intensities for the three CD4$^+$ subsets, classified by differential MHCII expression and cell size. The antibody expression frequencies described in the graphs are for the specific MHCII subsets. CD161$^+$ population is actually CD161 intermediary and therefore is not describing NK cells which should be CD161hi. pDCs are CD4$^+$ HIS24$^+$. No strain comparisons for the CD4$^+$ subsets showed any significant differences. f) Histograms describing anti-clec4b antibody staining in flow cytometer for either DA and $Clec4b^{E3}$ splenocytes. Three gates are shown in the T cells, B cells, and the CD4$^+$TCR- population. g) Mean fluorescence intensity for the Clec4b antibody illustrating expression per cell on the CD4$^+$TCR$^-$ population for both DA and $Clec4b^{E3}$.

marker His48 were expressed in noticeably differently patterns in different sub-populations (Fig 3C). When the expression of major histocompatibility complex type II molecules (MHCII) was analysed in combination with forward scatter (FSC), three distinct populations were detectable, each representing a third of the total CD4$^+$ pan T$^-$ population (Fig 3D). One population expressed high levels of MHCII, a second population expressed intermediate levels of MHCII, and a third population was found negative for MHCII. The MHCII$^{bright}$ population showed classical dendritic cell characteristics, such as high CD11b/c and CD172 (Sirpα) expression and negative for the granulocyte marker His48 (Fig 3E). The MHCII$^{dim}$ population, on the other hand, was granulocyte-like with very high expression of His48. The population also expressed His24, one of the markers for pDCs, here defined by CD45RA$^-$/TCR$^-$/His24$^+$/CD4$^+$ expression [17]. The MHCII$^-$ population was more monocyte-like, expressing CD68 and CD11b/c at levels comparable to the MHCII$^-$ population in CD4$^-$TCR$^-$, S1 Fig. iNOS expression was elevated in both MHC expressing populations (Fig 3E), from which one may infer that reactive nitrogen species are involved in shared regulatory functions. In conclusion, the $Clec4b$ expressing CD4$^+$ cells were shown to include two unique populations. The first is a dendritic cell-like population corresponding to the CD4$^+$ classical DC subset that has been previously described in the rat spleen [22], and which are similar to human cDC2. A second granulocyte-like population, which was undetected in the CD4$^-$/$Clec4b^-$ fraction and showed lower MHCII expression, corresponded to the pDC population. Since the pDC population was shown to express very low levels of $Clec4b$ (see Fig 2D), we conclude that the CD11b/c$^+$ DC population should be the main $Clec4b$ expressing CD4$^+$ population in the spleen. The production of a monoclonal antibody for rat Clec4b is described in recently published data from Daws et al. [18]. They also detailed how Clec4b is expressed mainly on CD4$^+$ DC/monocytes [18]. We found that Clec4b was only expressed in $Clec4b^{E3}$ CD4$^+$DCs but not in B or T cells (Fig 3F). Comparing cellular expression of the Clec4b receptor showed significant differential expression between DA and DA congenic rats, expressing $Clec4b^{E3}$ (Fig 3G).

### $Clec4b$ deficient DA rats showed enhanced expansion of CD4$^+$DCs at day 3 after oil injection.

With the intention of identifying how the non-sense mutations in the $Clec4b$ gene affect the cellular immune response during the early phase of experimental arthritis, splenocytes were harvested from the first days after oil-injection and then studied by flow cytometry. Assessing the cellular distribution in naïve spleens did not show any differences in the composition of cell sub-set populations between DA and $Clec4b^{E3}$ rats, S2 Fig. However, three days after intra-dermal injection of arthritogenic oil we observed an increase of CD4$^+$ myeloid cells in DA spleens, as measured *ex vivo* (Fig 4A). Furthermore, the MHCII expressing subset of CD4$^+$ DCs had expanded after activation and the difference between the two genotypes was more marked, discernible in the FACS plots (Fig 4B). After day three, MHCII expression was two-fold in DA compared to $Clec4b^{E3}$, and DA populations also showed enriched expansion of the adhesion molecule CD11b (Fig 4C). On day three no differences were seen for either T or B

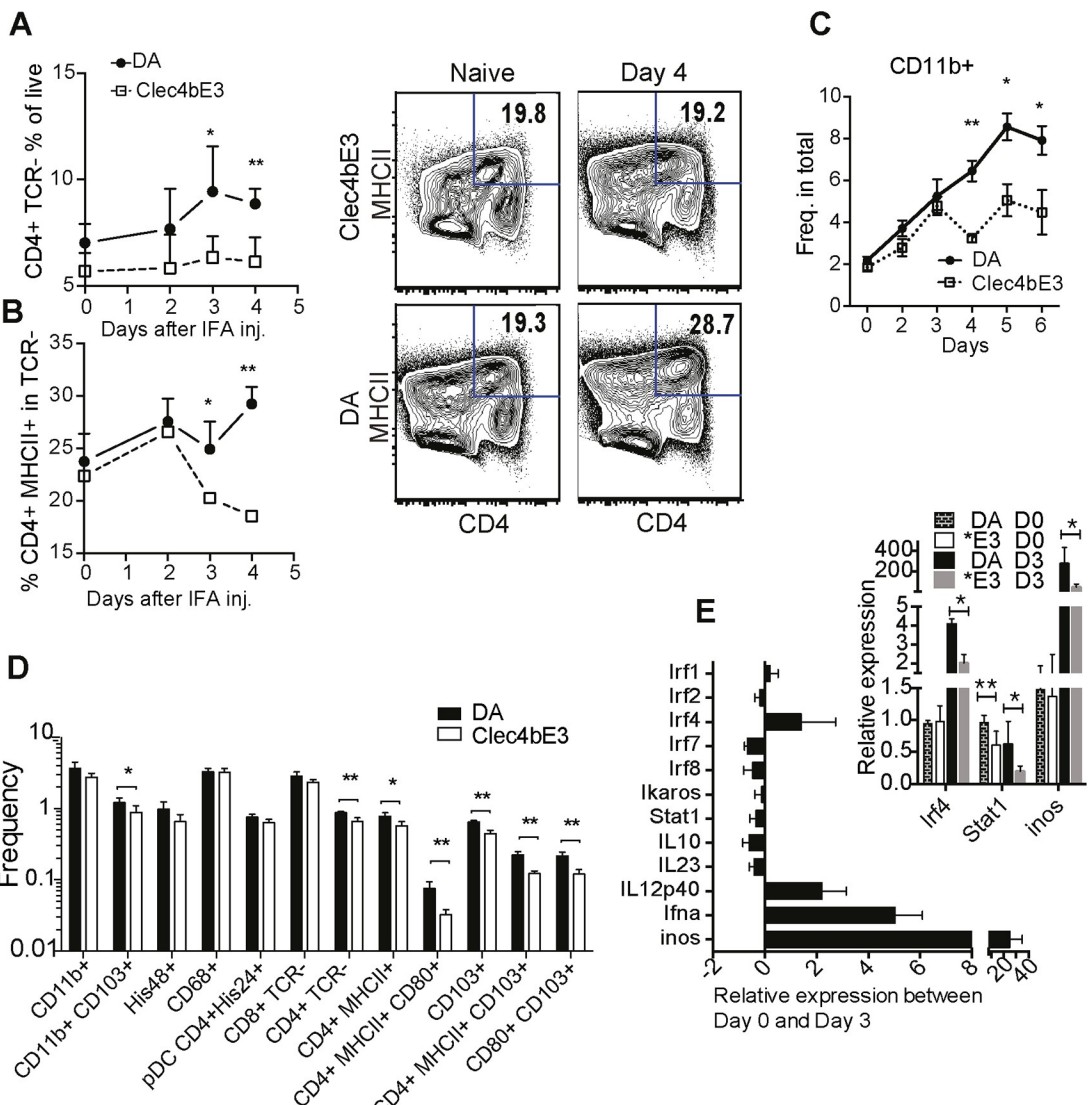

**Fig 4. *Clec4b* is expressed in naïve CD4⁺ MHCII⁺ DCs that expand greatly in *Clec4b* deficient rats after activation and initiate the development of inflammatory dendritic cells.** a) The kinetic distribution of CD4⁺ DCs (of TCR⁻ CD45+ cells) determined by flow cytometry in the spleen over time after IFA. n = 5 per group. Graph is a representation of 4 separate experiments. b) Kinetic distribution of geomean of fluorescence intensities of MHCII expression on CD4⁺ DCs over time after an intradermal IFA injection, and representative CD4 MHCII intensity plots for frequencies for samples from day 0, and day 4 after oil injection. Previously gated for TCR⁻ CD45⁺ cells. c) CD11b expression on TCR⁻ OX33⁻ MHCII⁺ cell as frequencies of total cells (*n* = 5 per group). d) The distribution of DC subsets in spleen day 3 after oil injection. The displayed distribution is for total cells and is a representative sample from 3 separate experiments (*n* = 5 per group). e) The relative expression of DC activation/differentiation regulating genes in the immune-precipitated CD4+ TCR- cells comparing expression day 0 to Day 3. Larger graph illustrates expression for both DA and *Clec4b*^E3. Data from the two genotypes were pooled to give a general overview. The smaller upper graph describes relative gene expression for genes that were differentially expressed between DA and *Clec4b*^E3. Significance stars depict Mann. Whitney Test * = *p<0.05*, ** = *p<0.01* bars are for SEM (*n* = 5 per group).

cell subsets in the spleen, S3 Fig. Populations of antigen-presenting cells were generally more activated in DA rats, so that there were more CD4⁺MHCII⁺CD80⁺ and CD4⁺MHCII⁺CD103⁺ cells (Fig 4D). However, neither pDC nor monocyte populations expanded differentially between DA or *Clec4b*^E3 during the first days of activation. Kinetic dissection of the early cellular responses to injection of an arthritogenic oil further supports dendritic cells as the cells initially involved in the *Clec4b* mediated responses and hence the arthritis development control.

Aiming to delineate the expression differences in the CD4$^+$ DCs, after oil injection, a set of DC-activation associated genes was compared by quantitative PCR between naïve cells and cells harvested three days after oil injection. CD4$^+$ DCs at day three had increased gene expression of *IL12p40*, *Ifna*, CD4$^+$DC activation marker *Ifr4*, and the pro-inflammatory regulator *Inos* suggesting an enhanced activated phenotype at day three. A comparison of DA to *Clec4b*$^{E3}$ expression showed significantly higher expression of both *Inos*, and *Irf4* in DA. Furthermore, there was already a significant difference in *Stat1* expression between DA and *Clec4b*$^{E3}$ in the naïve CD4$^+$ myeloid cell population (Fig 4E).

In summary, the DA CD4$^+$ DC population expanded at day 3 compared to rats with *Clec4b*$^{E3}$, displaying a distinct pro-inflammatory expression profile that promoted up-regulation and expansion of MHCII and adhesion markers such as CD11b. Thus, *Clec4b* deficiency lowers the threshold for classical CD4$^+$ DC activation and MHCII expression.

## *Clec4b* regulates activation of pro-inflammatory T cells before day five of OIA pathogenesis

Adjuvant induced arthritis models in the rat, such as PIA and OIA, are mediated by arthritogenic auto-reactive T cells activated a few days after adjuvant injection (see Fig 1C and 1D). Arthritogenic T cells can be isolated in draining lymph nodes as early as day 7 whereas the disease onset occurs around day 12 after adjuvant injection [23]. To investigate the *Clec4b*-mediated regulation of auto-reactive T cells during OIA development and to identify the earliest time point by which T cells can transfer disease, we transferred bead selected T cells (98% purity) from rats primed with mineral oil IFA into naïve recipient rats. Sufficient T cell activation for mild joint swelling was seen already at day 5 after oil injection (Fig 5A), whereas T cells from day 6 induced severe arthritis. Importantly, spleen T cells from DA but not DA. *Clec4b*$^{E3}$ induced arthritis in naïve DA recipient rats. However, the presence of *Clec4b* in the recipient was irrelevant. As *Clec4b* is expressed in DC and not on T cells, these data demonstrates that *Clec4b* suppress T cell activation before day 5 during the immune response. Moreover, the absence of T cell regulation in *Clec4b* sufficient recipient rats shows that this early regulatory effect was absent after the T cell activation has advanced to autoreactive. T cells from draining lymph nodes transfer arthritis similarly to T cells from spleen (Fig 5B).

Since the T cells isolated at day 5 after oil injection were already arthritogenic, we sought to assess the *in vivo* primed T cells from day 0 by flow cytometry. As T cells from both sources transfer arthritis T cells [23], subsets and activation markers were analyzed from both joint draining lymph nodes and splenocytes. Surprisingly, no expansion of CD4$^+$CD25$^+$ T cells was observed within the first 6 days after arthritis induction (Fig 5C). However, a dramatic decrease of CD62L$^+$CD4$^+$ T cells among all live cells was seen in the spleen, from 15% to almost undetectable numbers at day two. This dramatic reduction of naïve T cells did not differ between the genotypes (Fig 5C) but provides a clue that this population is downregulated immediately only hours after oil injection.

## Co-cultures of CD4$^+$ DCs and T cells identifies differential limiting effects on naïve T cell already 24hrs after stimulation

The CD4$^+$DC subset is able to efficiently present antigen to T cells, and induce T cell activation [21,24]. Since no clear population differences between DA and *Clec4b*$^{E3}$ rats could be identified for T cells *in vivo* within the first six days after disease induction, T cell activation was evaluated during *in vitro* stimulation where proliferative responses can also be studied. Co-cultures were made to assess the CD4$^+$DC regulatory function on T cell proliferation by labelling bead- selected T cells with CFSE. Moreover CD4$^+$DC have been shown to be superior in

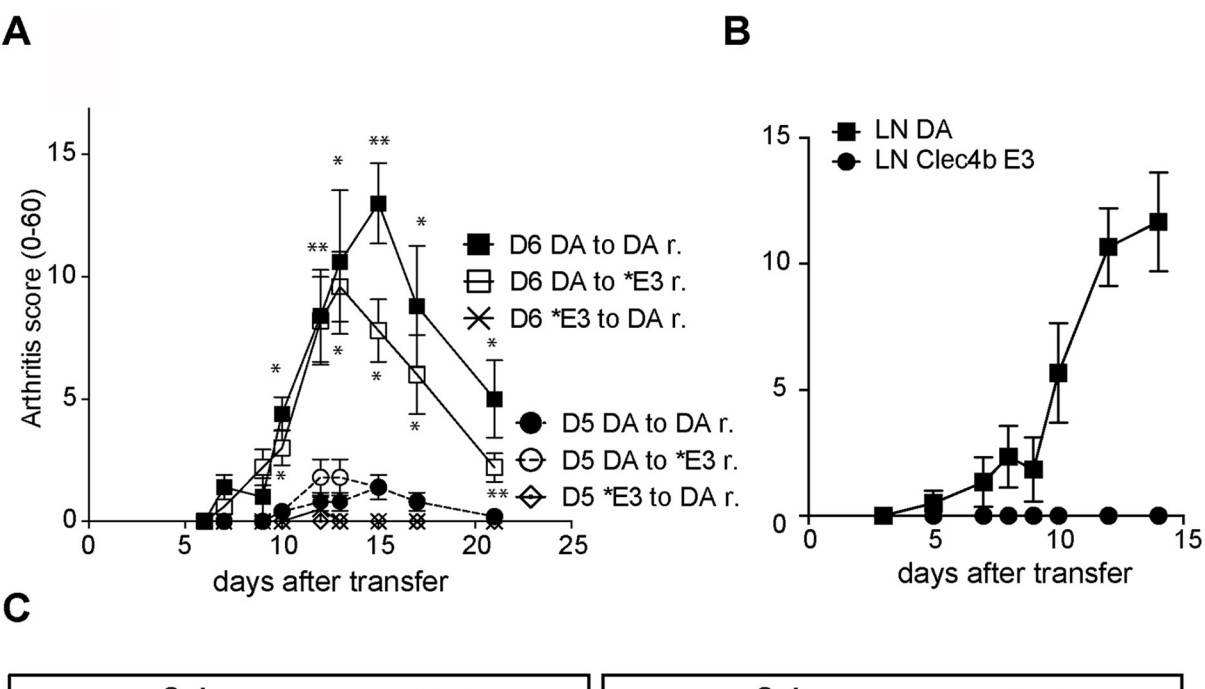

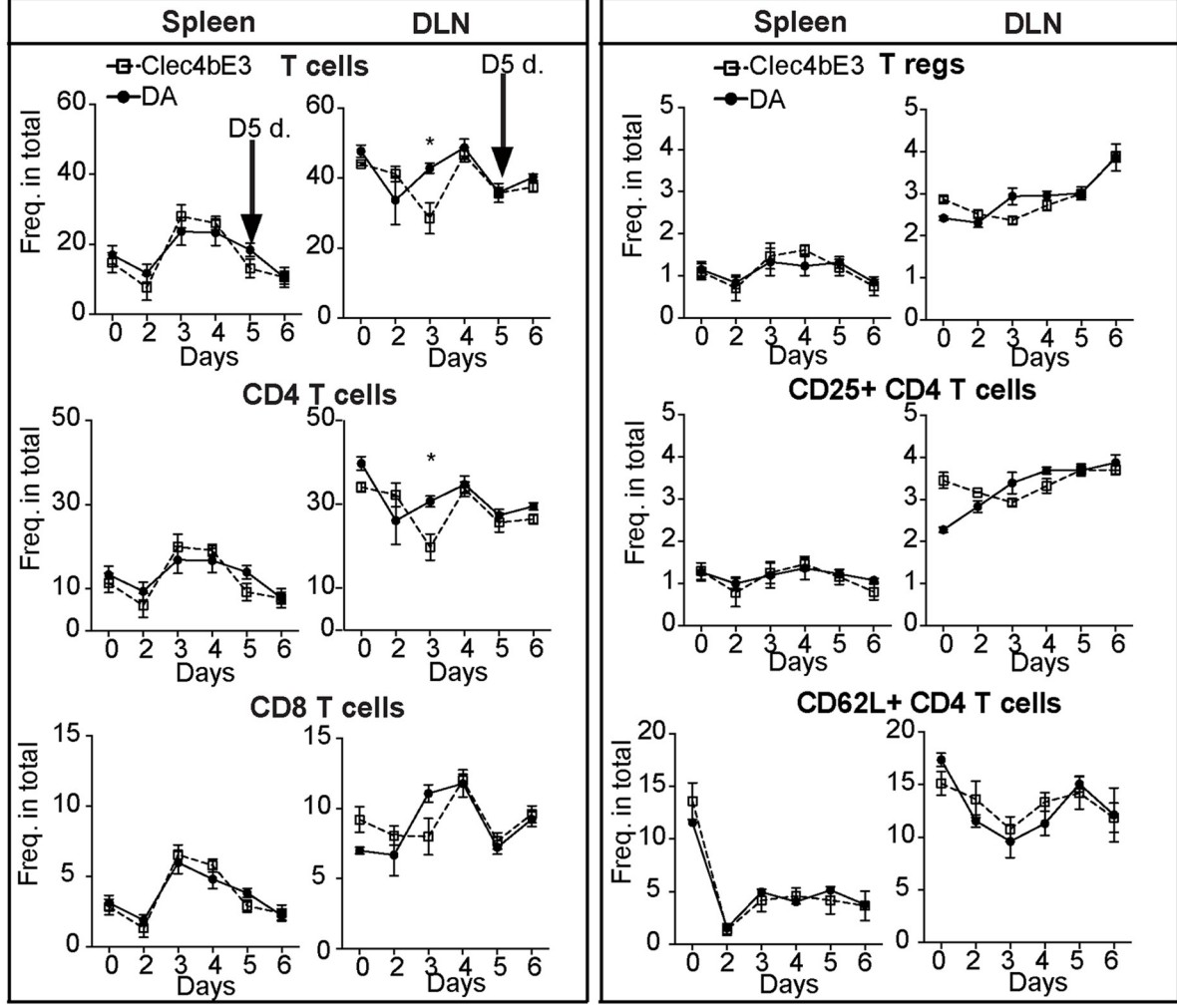

**Fig 5. Arthritis transmitting T cells are induced already 5 days after oil injection.** a) Transfer arthritis development in recipients after injection with 15 x $10^6$ spleen derived T cells from donor rats previously injected with mineral oil either 5 or 6 days beforehand. Donors and recipients are noted as d. and r. from either genotype; DA, or $Clec4b^{E3}$(*E3). $n = 5$ for all groups. b) Transfer arthritis development in DA recipients after injection with 15 x $10^6$ T cells from inguinal lymph nodes taken from donor rats day 6 after oil injection (both groups $n = 5$). c) Day to day distribution of T cell subsets as frequencies of total cell ($n$ per time point = 5). Statistical analysis was performed using Mann-Whitney Test $*$ = $p<0.05$ SEM. d) Day to day distribution of activation associated T cell markers as frequencies of total cells ($n$ per time point = 5).

stimulation of allogenic T cell responses in the rat compared to CD4$^-$DCs. We harvested allogenic T cells from rats with the MHC haplotype RT1$^u$, which differ from RT1$^a$ in the DA rat, from both naïve rats and day three after IFA injection. Syngenic DA T cells were also harvested from day three after IFA, from both Cle4b$^{E3}$ and DA rats. The different T cells were co-cultured with bead-selected CD4$^+$DCs from either DA or $Clec4b^{E3}$ in all combinations. Naïve allogenic T cells were co-cultured with naïve CD4$^+$DCs, and the day three derived T cells, either allogenic or syngenic, were all co-cultured with day three *in vivo* derived CD4$^+$DCs. Since arthritogenic T cells in the rat have been characterized as activated CD4$^+$ T cells, proliferation was assessed in CD25$^+$ CD4$^+$ T cells [23]. Proliferation was near 100% for both naïve and OIA day three harvested allogenic T cells however no difference in DC activation could be seen from *Clec4b*. This implies that *Clec4b* regulation on arthritogenic T cell development operates independent of MHCII. The CD4$^+$DC mediated T regulation occurs, most likely, by limiting bystander activation via cytokine secretion. Interestingly, syngeneic $Clec4b^{E3}$ derived T cells showed dramatically lower proliferation compared to DA. Since no effect could be seen for the two CD4$^+$DCs populations in the co-cultures, the restricted proliferation seen for $Clec4b^{E3}$ derived T cells must be due to regulatory differences during the first days *in vivo* rather than in the *in vitro* culture (Fig 6A and 6B).

In order to find an appropriate stimulatory agent triggering down-regulation of *Clec4b* expression, known pro-inflammatory compounds (ConA, LPS, mannan, thiodimycholate and CpGs) were incubated with freshly isolated naïve spleen cells. Remarkably, there was downregulation of *Clec4b* expression already after 24h of ConA stimulation (Fig 6C). However, there was no difference in viability of DCs after Con A between the two strains. Overnight ConA cultures activated naïve CD4$^+$DCs cells to become strong inducers of T cell proliferation compared with CD4$^-$DCs cells (Fig 6D). This ability to induce proliferation of CD4$^+$CD25$^+$ T cells was unique to DA derived CD4$^+$ DCs (Fig 6D).

Endeavoring to assess the restraining ability on T cell activation/proliferation from $Clec4b^{E3}$ derived DCs, two sets of DC:T cell co-cultures were prepared. Naïve Con A overnight activated CD4$^+$DCs from DA and $Clec4b^{E3}$ were combined with naïve T cells from either DA or $Clec4b^{E3}$. Overnight ConA stimulation of CD4$^+$DCs enabled the DCs from $Clec4b^{E3}$ derived rats to limit proliferation of CD25$^+$CD4$^+$ T cells from either DA and $Clec4b^{E3}$ equally efficiently after 72 hours of culture (Fig 6E).

In summary, $Clec4b^{E3}$ DCs from day 3 after oil injection failed to limit T cell activation in co-cultures with day 3 activated T cells, and T cell activation status correlated with the genotype of its origin (Fig 6B). Naïve $Clec4b^{E3}$ DCs, on the other hand, that had been ConA stimulated before co-culture with naïve T cells were able to inhibit activation and proliferation of T cells independent of the T cells *Clec4b* genotype (Fig 6E). These data reveal that the arthritis regulating gene *Clec4b* expressed in CD4$^+$ DCs can efficiently limit T cell activation in response to adjuvant/PRR stimulation, thus preventing the expansion of recently activated auto-reactive and potentially pathogenic T cells.

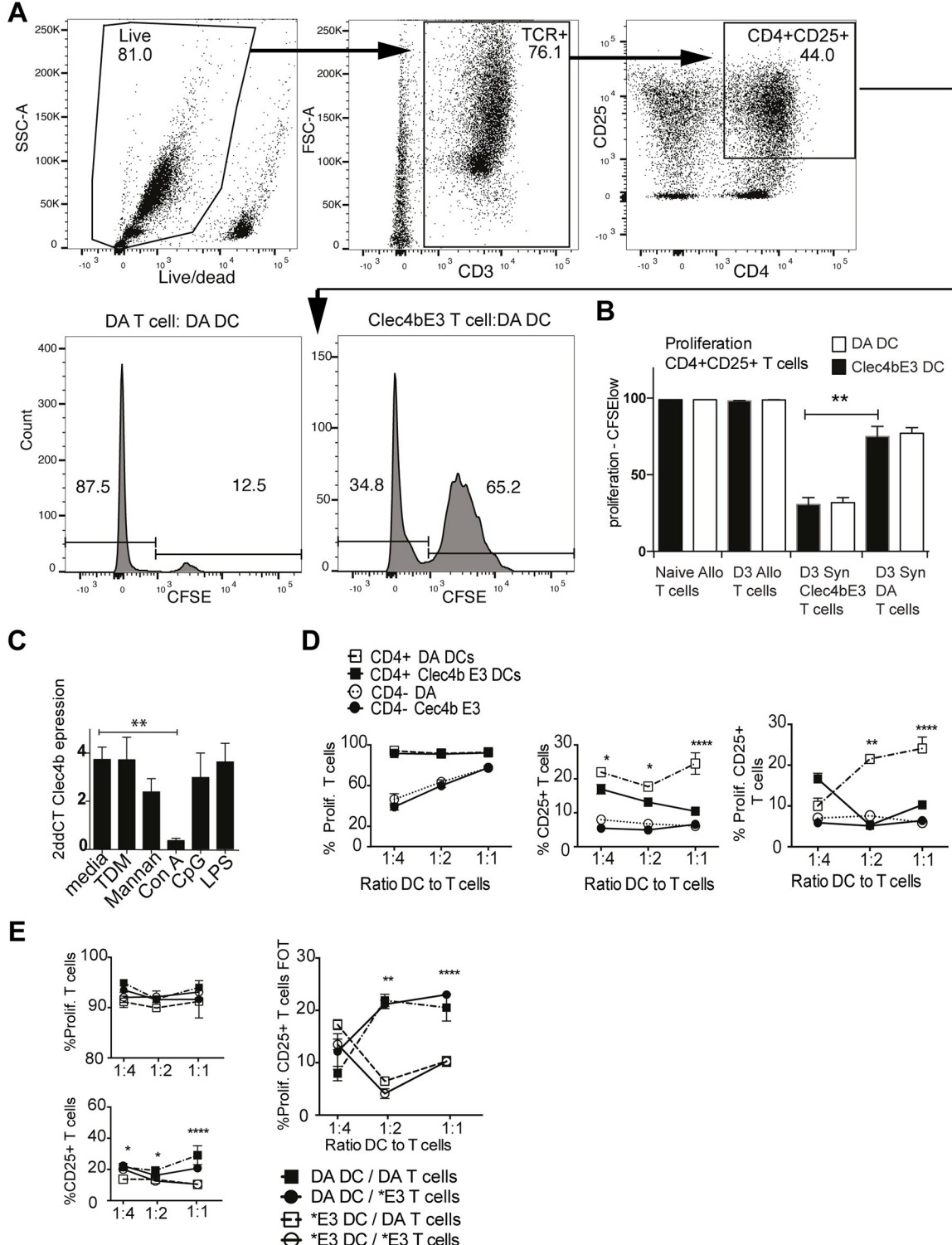

**Fig 6. *Clec4b* regulates CD4+ myeloid cell mediated suppression of naïve T cells in mixed lymphocyte reactions.** a) Illustration of the gating for activated T cells, by first gating for live cells, then T cells, and CD4+CD25+ cells followed by proliferation as carboxyfluorescein succinimidyl ester (CFSE) low. And an illustration of CFSE histogram for two samples. b) Graph describes proliferation as CFSE low labeled CD4+CD25+ T cells from a) for Naïve or OIA Day 3 activated allogenic T cells and OIA Day 3 activated syngenic DA or *Clec4b*E3 T cells. The T cells have been co-cultured with OIA day 3 *in vivo* activated DCs from both genotypes, or cultured with naïve DCs from both genotypes (only for naïve allogenic T cells). c) *In vitro* stimulated *Clec4b* gene expression in CD4+

TCR- spleen cells, after 24 h culture with either Con A, TDM, mannan, CpG, LPS, complete media only ($n = 5$ per group) Mann. Whitney Test ** = $p<0.01$ SEM. d) Comparisons of T cell subset in co-cultures mixing naïve DA T cells, cultured with Con A stimulated selected DCs, either CD4+DCs or CD4-DCs from DA or *Clec4b*[E3]. Frequencies of total cells are displayed for the co-culture at different DC:T cell ratios explained in 3 graphs; total CD4+ cells; total CD25+ CD4+ T cells; and proliferating (CFSE low) CD25+ CD4+ T cells. ($n = 10$ for all groups) Mann Whitney Test * = $p<0.05$, ** = $p<0.01$, *** = $p<0.001$ and SEM. e) Describes two four-way grouped co-culture experiments comparing the proliferative regulation from either DA or *E3 (*Clec4b*[E3]) CD4+ DCs on bead selected CFSE labeled T cells from both rat strains and mixed in combinations in co-cultures for 72 h. n = 10 for all groups. 3 Graphs illustrate proliferative T cell subsets: total proliferating T cells, total CD4+ CD25+ T cells and the frequencies of proliferating CD4+ CD25+ T cells in all cells. Graphs describes regulation of proliferation on recently harvested naïve T cells by *in vitro* Con A activated CD4+DCs in different T cell to DC ratios/concentrations. Mann. Whitney Test * = $p<0.05$, ** = $p<0.01$, *** = $p<0.001$ and SEM.

## *Clec4b* in recipient naïve DCs regulates T cell proliferation on day four activated T cells *in vivo* after challenge by innate stimulus and down-regulation of IL17 production

Given the fact that only DA.*Clec4b*[E3] and not DA derived T cells are prevented from the development into arthritogenic T cells after an intradermal injection of mineral oil, we propose that *Clec4b* should be able to allow DA donor T cell limiting responses *in vivo* after a second adjuvant injection in naïve recipients from the arthritis transfer model, as described in Fig 5A and 5B. Since T cells have already become arthritogenic 6 days after arthritogenic oil injection, we investigated the activation and proliferation status of T cells transferred earlier than this time point. CFSE+ labeled T cells harvested from day 3 after oil primed DA rats were transferred to naïve DA.*Clec4b*[E3] or DA rats. After transfering the primed T cells, the recipient was also challenged by an intradermal oil-injection to trigger the CD4+DCs. After 3 days in the recipients, the spleens were harvested and more than 30 million splenocytes were analyzed per sample by flow cytometer. 3 distinct population of T cells could be visualized in recipient spleens: an unlabeled population (CFSE-); a proliferative population (CFSE[low]); and a non-proliferating population (CFSE[high]) (Fig 7A). Whilst the non-proliferating CFSE[high] population corresponds to the main part of the donor T cell population, a considerable part of the donor CD4+ T cells (i.e. the CFSE[low] population) were proliferating. Intriguingly, all cells in the CFSE[low] subset were CD4+T cells, which corresponds well with previous findings from the Transfer arthritis model that has described the arthritogenic T cells as mainly CD4+[23]. Since these donor T cells most likely have been exposed to a form of CD4+DC bystander activation of abundantly polyclonal antigen specificity, there will have been no definite starting point for the T cell proliferation. Thus, the CFSE density could not be visualized into distinct cell divisions. Therefore, we used a second proliferation marker (Ki67) to assess proliferation within the CFSE categories. Further analysis of donor T cells in the *Clec4b*[E3] recipients revealed a significant regulation of the proliferating CD4+ T cells in terms of both cell expansion and activation (both lower KI67 and MHCII expression) (Fig 7B and 7C). Conversely, no difference in activation and proliferation markers was detected in non-dividing (CFSE[high]) and endogenous T cells (CFSE-) between both congenic rats. Furthermore, the effects of polyclonal stimulation of transferred cells indicate a role of *Clec4b* in modulating IL-17 producing T cells on the basis that a reduction in IL-17+ dividing T cells frequency was found in *Clec4b*[E3] compared to DA rats (Fig 7D and 7E). However, no changes in the frequency of IFNγ+ T cells was detected. Notably, CFSE[high] population failed to produce both IL-17 and IFNγ while endogenous T cells (CFSE-) produced IFNγ only. These findings corroborate the role of *Clec4b* in limiting T cell activation and proliferation *in vivo*. They also infer that *Clec4b* plays a part in controlling Th17 cells that together account for a substantial proportion of the DA susceptibility to adjuvant triggers that lead to arthritis development in the rat.

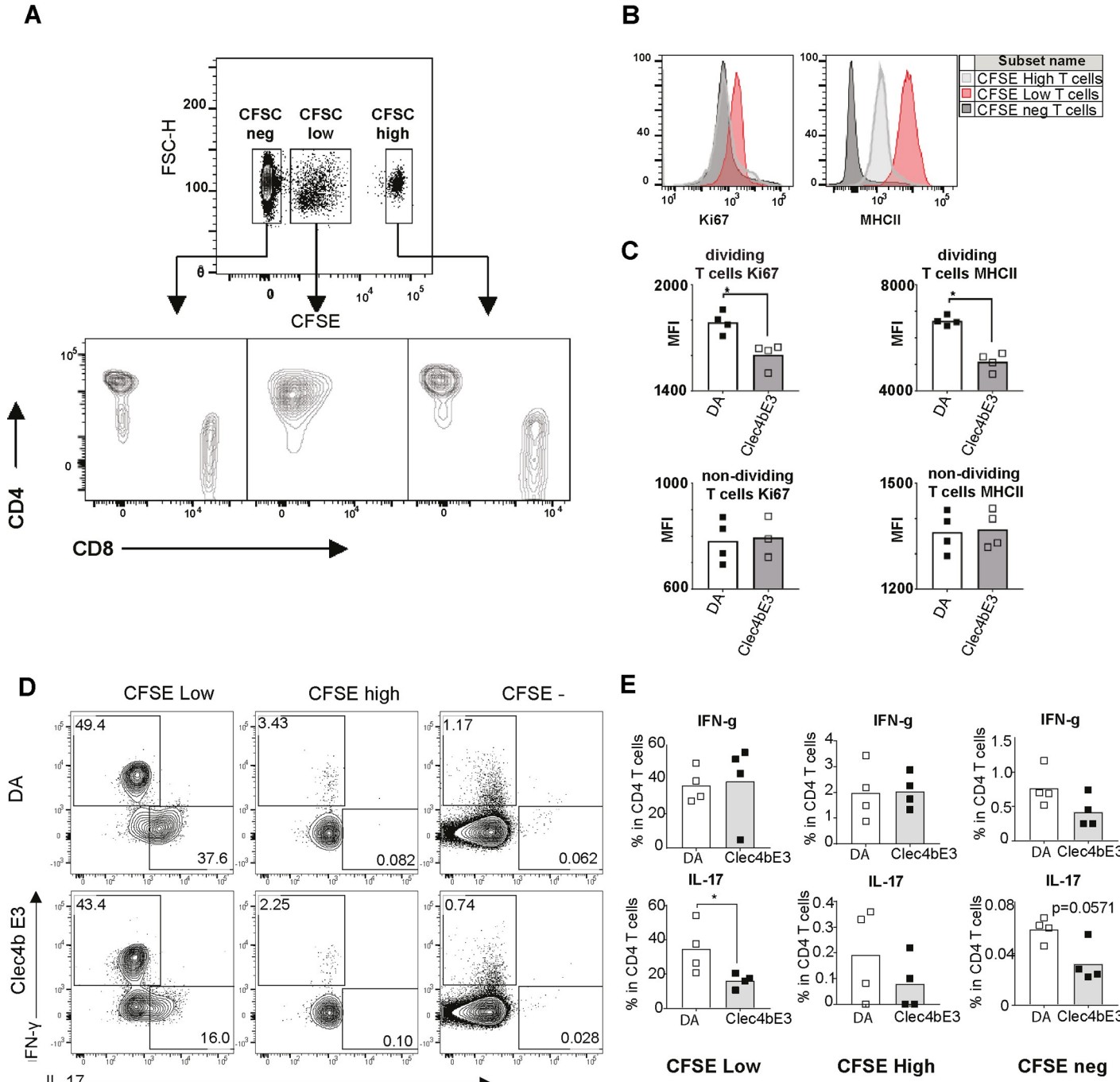

**Fig 7. _Clec4b_ regulate T cell proliferation also _in vivo_ and the activation correlate with down-regulation of IL17 production.** DA rats were injected intradermally with oil and 4 days later the spleens were used as a source of oil primed T cells. These were CSFE labeled and injected intravenously into naïve DA and _Clec4b_[E3] recipients that had been injected intradermaly with mineral oil the same day. a) Proliferation FACS plot describing CFSE intensity in CD4 versus CD8 T cells harvested from spleens of the recipients day 3 after oil injection. CFSE negative T cells are the recipient endogenous T cells. The CFSE[high] are the non-proliferating T cells from the donors and the CFSE[low] are the donor T cells that have been proliferating in the recipient. b) Histogram over the distribution of the three groups from a and how they differ in the expression of both Ki67 and MHCII on their surface. c) Graphs describing how the two donor groups, non-dividing and dividing T cells, differ in the expression of MHCII and Ki67 between DA and _Clec4b_[E3] recipients. d,e) Illustration on how the expression of the two cytokines IFNg and IL17 is distributed between the three groups of differentially proliferating T cells between the Clec4b sufficient and deficient recipient rats. d) Illustrative FACS plot. e) Statistics analysis and graph. n = 4 for all groups, Mann Whitney Test * = $p < 0.05$, ** = $p < 0.01$, *** = $p < 0.001$ and SEM.

## Discussion

The identification of a single nucleotide polymorphism controlling the expression of *Clec4b* in the rat highlights a new role of *Clec4b* in regulation of autoreactive T cell activation. The suggested identification of the *Clec4b* SNP to be causative was based different type of evidence. Firstly, it was found within a 200kb chromosomal interval defining the QTL Pia7. This genomic region harbors three protein coding genes, whereof two had non-synonymous SNPs. Of these, only the SNP in *Clec4b* showed functional evidence as it led to differential gene expression in naïve rats. Secondly, data from several genetic mapping studies aiming to identify arthritis susceptibility regulating loci in the rat, shows that only the DA version of Clec4b segregates with disease susceptibility, and not strains lacking the *Clec4b* SNP but maintaining the polymorphism outside the *Clec4b* locus. Thirdly, only the DA *Clec4b.E3* congenic, and not the DA rat, expresses the Clec4b protein. More specifically, Clec4b is highly expressed on CD4 + DCs, a population known to be regulating MHCII interaction and T cell activation. Interestingly after an intradermal oil injection, the DA strain displays a rapid onset of autoreactive T cells whereas *Clec4b.E3* rats have normal control of T cell activation after oil injection. *Clec4b* is a C-type lectin receptor, expressed in classical CD4$^+$DCs, and we now find that it is essential for the control of bystander activation of autoreactive T cells during immune priming. Deficient expression of *Clec4b* allows the escape and expansion of autoreactive T cells leading to arthritis.

T cells from a rat strain deficient in *Clec4b* expression are able to transfer arthritis to naïve rats already 5 days after being activated by adjuvants. Two-photon imaging of DC T cell interactions *in vivo* has revealed that, after adjuvant stimulation, T cells quickly attach to the activated DCs via interaction with peptide-loaded MHCII that then leads to T cell viability decisions within the first 12 hours [25]. Our data suggest that *Clec4b* is a regulator of these viability decisions and thereby helps to prevent T cells from responding to self-peptides that are already present on classical DCs and vulnerable to activation. It is suggested that immature/steady state DCs have a tolerogenic function during steady-state [26]. During pattern recognition, receptor-mediated activation of DCs will occur at a critical time point when these self-presenting DCs need to differentiate or die in order to prevent development of autoimmunity [27]. We suggest that Clec4b function is a control switch for the tolerogenic DCs to either transform or die during the short interval when the DCs have received secondary inflammatory signals, yet are still capable of activating T cells directed to an immunogen that are, in most cases, derived form a pathogenic intruder. We have identified two triggers of Clec4b, mineral oil and Con A, which will activate DCs lacking Clec4b expression, leading to uncontrolled T cell proliferation. The presence of the lectin receptor Clec4b/Dcar on the DC surface cause a more controlled activation of DC with Clec4b$^{E3}$, compared with DC with Clec4b$^{DA}$, inducing less expression of proinflammatory genes such as Stat1, Irf4 and iNos, that could limit bystander T cell activation.

Importantly, mature T cells that possess a variable degree of self-reactivity and have escaped negative selection—through low avidity or by the absence of the relevant tissue antigens during their selection—can potentially cause autoimmune disease [28,29]. During the first priming immune response there are multiple mechanisms leading to activation of T cells with high affinity to peptides derived from the invading pathogen or immunogen. In the same time period, however, other cells with variable degree of self-reactivity need to be suppressed. We have now identified a way in which such autoreactive T cells can be suppressed by dendritic cells expressing Clec4b during a limited period during the immune priming. If Clec4b is not functioning properly, autoreactive T cells could be activated that could be pathogenic and cause an autoimmune arthritis that mimics rheumatoid arthritis.

The CD4[+] DCs population in the rat spleen have previously been described to be highly efficient inducers of MHCII restricted T cell activation and to express enhanced levels of co-stimulatory receptors [22]. These also express PRRs on their surface [21]. A possible mouse counterpart to the rat CD4[+] DC is CD11b[+] CD8a[-]DC, which have been described as a mediator of peripheral tolerance via MHCII [30]. Targeting antigen bound antibodies to a steady-state associated C-type lectin receptor DEC-205 without antigen can induce tolerance in mice [31, 32]. In the mouse, there are two isoforms of *Clec4b* and of these *Cle4b2* could have a different function as it is expressed on CD8[+] DC and involved with cross-presentation to MHCI of peptides circulating in the early endosome [13]. Instead, the less homologous isotype *Clec4b1* (DCAR2) could be functionally more similar to rat Clec4b as it is co-expressed with *Clec4a4* (DCIR2) [33] on mouse CD8[-] CD11b[+] DCs, the mouse splenic DC subset analogous to the rat CD4[+] DCs. *Clec4a4* was recently identified as a mediator of inflammation and MHCII:T cell immunity because both its function and the DC specificity show similarities with the identified function of rat *Clec4b* [14,34]. However the proportion of CD4[+] and CD8[+] dendritic cell populations differ between rats and mice where rat DCs better resembles human DC subsets [35]. However, rat Clec4b most likely does not have a completely overlapping set of functions with human CLEC4C, since the latter is a human pDC bio-marker [36,37] and Clec4b fails to show significant expression in rat pDCs.

*Clec4b* expressing CD4[+] DCs plays an important role in limiting the expansion of autoreactive T cells during an immune response and, in particular, when the immune response is triggered in the absence of strong antigens. This new role of antigen-presenting DCs is of fundamental importance in understanding autoimmune diseases but may also have important implications for current explanations of the lack of autoreactive T cell responses in growing tumors.

## Methods

### Selective breeding of congenic animals

DA/ZtmRhd (DA.RT1[Av1]) and E3/ZtmRhd (E3) were originally obtained from Zentralinstitut fur Versuchstierkunde (Hannover, Germany) but maintained as inbred strains in our laboratory for more than 20 generations. The original 2.3Mb DA.E3-*Pia4* congenic strain was produced and back-crossed to DA for at least 16 generations[11]. Two recombinant congenic strains were produced by marker-assisted selective breeding from back-crossed DA.E3-*Pia4* heterozygotic parents in an F2 inter-cross. During the process of producing the recombinant strains the congenic strains were back-crossed to DA three more times. The DA.E3-*Clec4a* recombinant was obtained after 567 meiosis and occurred between MSaplec710 and MSaplec740 (which is downstream of *Clec4a* before *Clec4b2*. DA.E3-*Clec4bde*) was produced from intercrossed DA.E3-*Pia4* in two steps. The first recombination occurred between MSaplec573 and MSaplec633 upstream of *Clec4a* and happened after 1723 meiosis. The second recombination in the DA.E3-*Clec4bde* fragment occurred after 4831 meiosis between D4Rat90 and MSaplec1003 right after the *Clec4e gene*. The congenic fragment DA.E3-*Clec4bde* is a 340 kb congenic fragment between164.61 Mb to 164.95 Mb on chromosome 4 conformed to the Baylor Rn4 2004 assembly and is illustrated in the schematic in Fig 1A. Markers for the production of the two subcongenic strains were: From the distal top D4MIR55F GCTGAGTTTTGG GGTTGATTT, D4MIR55R GGTACAGCCGCCTTCTT, D4WOX49F TGATCTCAGCACCC ATGTAA, D4WOX49R GCAGGACAGCCAGGACTAC, MSaplec573F ACCAGTAAACACC ATCATTGG, MSaplec573R CACATTGATGAAAGGCACAA, MSaplec637F CACTTCCCA GACAAAGTTGC, MSaplec637F AGGTACAATGGCACACACCT, MSaplec710F CTTCT ATTACTTAACATCATTGCCATA, MSaplec710R GGGTCAAACTGCCAGATAAA,

MSaplec740F AGTAATGCCCCATGTGAAAA MSaplec740R GACCTGCATGCACATGAA
TA D4RAt90F CTCAGCATGCTCGCCACT, D4RAt90R TGTTATGACACTGTCTTATG
GTGAA, MSaplec981F TGCACCTGAGCATGTAAGTGTAT, MSaplec981R CCTGAACAG
TCCTGACATAAACC.

Allogenic rats were congenic DA.RT1$^u$. Animals were maintained at the Scheele Labora-
tory, in the Karolinska Institutet, in a pathogen-free environment according to the Federation
of European Animal Laboratory Science Association guidelines (FELASA). All procedures and
animal experiments have been approved by the local ethics committee (Stockholms djurför-
söksetiska nämd). The animals were maintained in a 14h light and 10h dark cycles and held
individually in cages containing wood shavings and aspen. Moreover, they had free access to
water and were fed with the standard rodent chow.

### Arthritis experiment

Arthritis was elicited by a single intradermal injection at the dorsal side of the tail base. For
PIA induction, 150 μl pristane (2,6,10,14-tetramethylpentadecane, 95%; Acros Organics, Mor-
ris Plains, NJ, USA) was injected. For induction of OIA, 200 μl mineral oil (incomplete
Freund's adjuvant /IFA/, BD, Franklin Lakes, NJ, USA) was injected. For induction of CIA, rat
collagen type II (CII), dissolved in 0.1$M$ acetic acid and emulsified in IFA, was injected. All
used rats were >8 weeks old. All experiments were performed with littermate rats that were
age-matched, distributed within the cages and blind evaluated by the investigator. The rats
were regularly inspected, including monitoring the limbs for arthritis development by a previ-
ously described macroscopic scoring systems. Briefly, 1 point was given for each individual
swollen and erythematic toe and up to 5 points for an inflamed ankle (15 points in total per
paw) [38,39]. The scoring was carried out every second or third day for 20–30 days after dis-
ease induction. No points were given to deformed paws that did not exhibit signs of
inflammation.

For transfer OIA, we harvested spleens or inguinal lymph nodes. Concisely, at day 6 or day
5 post injection organs were made in to single cell suspension. T cells were positively selected
using magnetic CD6 Macs beads. Cells were then culture for 56 hours in complete media and
stimulated with 10 μg/ml αTCR (R73) and 5 μg/ml αCD28 (JJ319) and later injected into
recipient naïve rats. Cells from DA or $Clec4b^{E3}$ were injected into DA or $Clec4b^{E3}$ recipients
and arthritis development was monitored using a 0–60 macroscopic scoring system. All exper-
iments follow established guidelines involving littermate controls, mixing of groups in cages
and blinded scoring evaluation.

### Cell preparation and immuno-sorting

Splenocytes were harvested from naïve spleens 1–5 days after oil injection. Single cell suspen-
sions were prepared as follows. Inguinal joint draining LNs and spleens were harvested from
euthanized rats and digested for 30 min at 37°C with 1 ml digestion buffer (1 mg/ml collage-
nase IV, Sigma-Aldrich, St. Louis, MO, USA) in DMEM media (Sigma-Aldrich), Spleens were
passed through 40-μm cell strainers and washed twice in sterile PBS. Red blood cells were
lysed by adding 2 ml of ACK buffer (155 mM $NH_4Cl$, 10 mM $KHCO_3$ and 0.1 mM EDTA) to
the pellet. After incubation for 3 mins at 4–8°C, lysis reaction was stopped by adding cold PBS,
centrifuged (8 min, 300$g$) and washed in ice-cold PBS.

Immuno-precipitation of samples in single cell suspensions was performed according to
the manufacturer's protocol (Miltenyi biotec. Gladbach, Germany). T cells were selected using
OX52 (CD6) Macs beads. T cell negative fractions were harvested from the flow-through. The
T cell-purity was never less than 97%, the non-T cell flow-through had less than 3%

contaminant T cells. The non-T cells flow-through was used for sorting the CD4+ non-T cell fraction using Macs CD4+ beads (coated with OX38). CD8+ or CD11b/c+ non-T cells were selected by using streptavidin-coated Macs beads coupled to biotin conjugated anti-CD8 anti-bodies (clone OX8, BD Franklin Lakes, NJ, USA) or CD11b/c+ (clone OX42, Biolegend San Diego, CA, USA). B cells were selected from single cell suspensions using CD45RA coated Macs beads (Clone OX33). Neutrophils were selected from EDTA-treated whole blood. First from this selection were PBMCs isolated by 6% dextran and centrifugation, followed by neutrophils isolated using Ficoll Hypaque, according to *Current Protocols in Immunology 7.23.1*.

## Gene expression

RNA was extracted from -80C snap frozen tissue samples using RNAeasy (Qiagen, Hilden Germany). cDNA was prepared by reverse transcriptase (iScript Bio-Rad Hercules, CA, USA) and qPCR amplification was performed using iq-SYBR green supermix (Bio-Rad Hercules, CA, USA). To eliminate potential splicing effects two different primer sets that cover different regions of the *Clec4b* gene were used in the tissue and cell type-characterizing experiment. The amplified sequence was cloned and sequenced for both primer sets and both sets were verified to have amplified *Clec4b* specific sequences and neither of the highly similar other APLEC encoded receptors. All other genes show gene expression from 1 primer set amplicon.

Clec4b2:1F TGCTCATCTGTTGGTGATCCA, Clec4b2:1R TGTAAAATAACCCCAACG AGTGTCTA, Clec4b2:2F AAAACTGCCCCAAGGTAAGG, Clec4b2:2R GAAGCAGGTGC TGAGGAGTAA, Clec4e.F TTTCACAGAGTCCCTGAGCTTCT, Clec4e.R TCCCTCATGGT GGCACAGT, Clec4d.F CACAAGGCTAACATGCATCCTAGA, Clec4d.R GCAAAGTAACA GTTAGACTGGAATGCT, Clec4a1F CATTCGTCCGTGGAAGACAAA, Clec4a1R TGCAG AGTCCCTGGAAGT, Clec4aF CCATAGCAAGGAAGAACAGGACTT, Clec4aR TGAATCC CAGAGCCCTATAAAATAA, Clec4a2F TCCTTGGGCACCTACTCAGAA, Clec4a2R GG AACCTGGCTTACTTGGATGA,Clec4a3F AGCCAGGAAGAGCAGGATTTC, Clec4a3R TGATCGACCCATTGCCATT IRF1F CCATTCACACAGGCCGATAC, IRF1R TCCTCG ATGTCTGGTAGGGA, IRF2F ACAACACTTACACCTTGCGG, IRF2R GCATGGTAC CCTCTCAGTGT, IRF4F GCCAACCCAGGTTCATAACT, IRF4R TGACTGGTCAGGGGC ATAAT, IRF7F TCTGGATGAAGCTGATGCAC, IRF7R CCTCTGGGGGTGGTAAGTT, IRF8F CCCGAGGAAGAGCAAAAAA, IRF8R TCAGCTCCTCGATCTCTGAA, IkarosF GCCACAACTACTTGGA, IkarosR GCTGTGGTTAGTTTCTTCCTTAATGA, IL10F CCCTCTGGATACAGCTGCG, IL10R GCTCCAGTCCCTTGCTTTTATT, IL23_F AAA AGTGACGTGCCCCGTAT, IL23R GCAAACAGAACTGGCTGTTG, IL12p40F AGCAGCA GTTCCCCTGAGTCT, IL12p40R GGCACGCCACTGAGTACTTCT, Stat1F CCACAAC TGTCTGAAGGAGGA, Stat1R TCACGGTGTTCTGAATATTTCC, IFNaF CAGCAGATC CTCAGCCTCTT, IFNaR TGCTGCTGGAGGTCATTACA, Nos2F GGAGCGAGTTGTGG ATTGTT, Nos2R GGGAAGCCTCTTGTCTTTGA, GapdhF TCAACTACATGGTCTACA TGTTCCAG, GapdhR TCCCATTCTCAGCCTTGACTG, B2MF CGTGATCTTTCTG GTGCTTGTC, B2MR TTCTGAATGGCAAGCACGAC, TBPF AGAACAATCCAGACTAG CAGCA, TBPR GGGAACTTCACATCACAGCTC, PpiaF ATTCATGTGCCAGGGTGGTG, PpiaR GGACCTGTATGCTTCAGGATG,

All quantifications were based on the Livak method 2-ddCT and minimum one of 3 house-keeping genes PPIA, Gapdh, TBP, or beta-2-macro.

## Flow-cytometry

Cells in single-cell suspension were added to 96-well V-bottom polypropylene plates (BD Falcon) at $10^6$ cells/well and incubated with saturating concentrations of mAbs (see below). Cells

were washed with FACS buffer and incubated in BD Cytofix/Cytoperm for 20 mins at room temperature and washed twice in BD Perm/Wash before staining with Abs to CD68. Fluorescence minus one control were used in all experiments. Dead cells were stained with Live/Dead Violet (Invitrogen) after washing the cells twice in Dulbecco's PBS without sodium azide and FCS. A SORP BD LSR II analytic flow cytometer (BD Biosciences) was used for acquisition and the data were analyzed with FlowJo (Tree Star). The following Alexa Fluor 488–, PE-, PerCP-Cy5.5–, allophycocyanin-, allophycocyanin-Cy7–, Qdot-655–, and biotin-conjugated Abs were used for flow cytometry: 2.4G2/BD Fc Block (CD32), OX-8 (CD8a), OX-1 (CD-45), HRL-1 (CD62l) and WT.5 (CD11b) were purchased from BD Pharmingen (San Diego, CA); R73 (αβTCR), OX-62 (CD103), W3/25 (CD4), 24F (CD86), 3H5 (CD80) purchased from Bio-Legend (San Diego, CA USA); OX-39 (CD25), OX-17 (RT1-D), His24 (CD45R/B220), His36 (ED-2) were purchased from eBioscience (San Diego, CA); and ED1 (CD68), purchased from AbD Serotec.

## Co-culture assay

Splenocytes from either naïve- or oil-injected rats were prepared in single cell suspension and selected with antibody coated beads, either in a negative or positive selection for the following surface receptors: CD4+ CD6- (CD4+ DCs), CD4-CD6- (CD4- DCs) or CD6+ (total T cells) as previously described. Cells were cultured in complete DMEM media (10% FCS, 1M Hepes (Sigma, St Louis, MO, USA), 50 uM 100U/ml penicillin, 100 ug/ml streptomycin, 50 uM β-mercaptoethanol (Sigma St Louis, MO, USA)). *In-vitro* stimulation of naïve CD4+ DCs was prepared in 24-hour cultures of complete media in addition to one of the following: 3ug/ml Concanavalin A (Sigma, St Louis, MO, USA); 5 ug/ml Trehalose Dimycolate (TDM)(Sigma, St Louis, MO, USA); 10 ug/ml Mannan (Sigma, St Louis, MO, USA); 50 ug/ml CpG-A 1668 (Invivogen, San Diego, CA, USA) or 3 ug/ml LPS (Sigma, St Louis, MO, USA). For proliferation assays magnetic-bead selected T cells were labeled with CFSE 2 uM before 72 hrs culture in complete media with either *in vivo* activated CD4+ DCs taken from day one, day two or day three of after oil injection or CD4+ DCs activated *in vitro* for 24 hours with Con A.

## *In vivo* proliferation assay and cytokine ELISA

T cells were harvested from four DA rats at day four after oil injection. T cells were harvested from whole spleen and lymph nodes and selected using magnetic CD6 Macs beads and subsequently labeled with CFSE. The labeled T cells where then immediately injected to naïve rats from either the DA or *Clec4b*[E3] strain. T cells from one rat from day four oil injected donors, were split between one DA and one *Clec4b*[E3] recipient. The naïve recipients where then challenged with an intradermal injection of oil and, three days later, spleens and draining lymph nodes were collected. The cells were labeled with antibodies to CD3, MHCII and CD25 in order to measure expression together with CFSE from the donor T cells. More than 100 million cells where assessed per recipient animal using a flow cytometer. Other cells were cultured with PMA, ionomycin and brefeldin A for 6 hours and further stained intracellularly with anti-IL17, IFNg and IL10 antibodies and analyzed with the flow cytometer.

## Ethics statement

All animal experiments were approved by the local ethics committee (Stockholms djurförsöksetiska nämd) and all experiments were carried out according to method approved (Ethical Approval Number N35/16). Animal anesthesia was performed using an inhalant gas mixture of isoflurane and air. All euthanasia was carried out using $CO^2$ gas.

## Supporting information

**S1 Fig. Illustration of cell population subsets within both the CD4+ as well as the CD4-fraction of antigen presenting cells depending on their expression of MHCII.** Three spleens harvested from day 3 after oil injection were selected first for T cells were the negative fraction was collected and selected a second time for CD4 expression. The twice selected cells were either CD4$^+$ or CD4$^-$, each represented as 3 samples. The samples were then labeled with fluorescence conjugated antibodies to identify subsets of cells. Since there appeared to be 3 clear subsets of CD4+ DCs depending on the MHCII expression the cells were first gated as either MHCII negative, MHCII high/bright and MHC dim. The value on the x axis is the present of total number of cells.
(TIF)

**S2 Fig. Illustration of cell subsets in the naïve spleen of DA versus *Clec4b*$^{E3}$.**
(TIF)

**S3 Fig. Illustration of distribution of B and T cell subset in the spleen of DA versus *Clec4b*$^{E3}$ at 3 days after oil injection.**
(TIF)

**S1 Data. Excel-file that includes data to all graphs for the figures in this manuscript.** The file have 1 sheet per figure where the different subset graphs in each figure are named by their index.
(XLSX)

## Acknowledgments

We thank Michael Daws and Erik Dissen for providing the Clec4b antibody. We also thank our animal house technicians Carlos and Kristina Palestro for their tremendous help in maintaining the animals.

## Author Contributions

**Conceptualization:** Liselotte Bäckdahl, Ulrika Norin, Rikard Holmdahl.

**Data curation:** Liselotte Bäckdahl, Ulrika Norin.

**Formal analysis:** Liselotte Bäckdahl, Mike Aoun.

**Funding acquisition:** Liselotte Bäckdahl, Rikard Holmdahl.

**Investigation:** Liselotte Bäckdahl, Mike Aoun, Ulrika Norin.

**Methodology:** Liselotte Bäckdahl, Mike Aoun, Ulrika Norin, Rikard Holmdahl.

**Project administration:** Liselotte Bäckdahl, Rikard Holmdahl.

**Resources:** Rikard Holmdahl.

**Supervision:** Liselotte Bäckdahl, Rikard Holmdahl.

**Validation:** Liselotte Bäckdahl, Mike Aoun.

**Visualization:** Liselotte Bäckdahl.

**Writing – original draft:** Liselotte Bäckdahl.

**Writing – review & editing:** Liselotte Bäckdahl, Mike Aoun, Rikard Holmdahl.

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
