## [Decision Letter · Decision Letter 0]

14 Sep 2019

Dear Dr Holmdahl,

Thank you very much for submitting your Research Article entitled 'Positional cloning of Clec4b - a novel regulator of bystander activation of auto-reactive T cells and autoimmune disease' to PLOS Genetics. Your manuscript was fully evaluated at the editorial level and by independent peer reviewers. The reviewers appreciated the attention to an important problem, but raised some substantial concerns about the current manuscript that need to be addressed. Based on the reviews, we will not be able to accept this version of the manuscript, but we would be willing to review again a much-revised version. We cannot, of course, promise publication at that time.

If you decide to revise the manuscript for further consideration at PLOS Genetics, please aim to resubmit within the next 60 days, unless it will take extra time to address the concerns of the reviewers, in which case we would appreciate an expected resubmission date by email to plosgenetics@plos.org.

[LINK]

We are sorry that we cannot be more positive about your manuscript at this stage. Please do not hesitate to contact us if you have any concerns or questions.

Yours sincerely,

Derry C. Roopenian

Associate Editor

PLOS Genetics

Gregory Barsh

Editor-in-Chief

PLOS Genetics

Reviewer's Responses to Questions

**Comments to the Authors:**

Reviewer #1: Bäckdahl et al perform positional cloning and suggest a polymorphism in Clec4b confers resistance to arthritis in rats. They find that Clec4b is primarily expressed in CD4+ dendritic cells (DCs) and that mice carrying the susceptibility allele show an increased activation of CD4+ DCs upon stimulation. They then attempt to demonstrate, using both in vitro and in vivo assays, that the resistance allele in DCs suppresses T cell activation.

I have three major concerns:

1 – The authors never perform a complementation assay to prove that Clec4b is responsible for the phenotypes observed. This needs to be addressed experimentally.

2 – Although the authors present various in vitro and in vivo assays, it is not clear that the biological effects observed are intrinsic to DCs. This can be addressed with the following series of in vitro experiments. (1) T cell responses from both strains of mice can be stimulated in vitro. This will allow to unequivocally determine whether the alleles impact T cell responses. (2) DCs from both strains should be cultured in vitro with naïve allogeneic T cells. This will allow to determine if the alleles expressed in DCs impact T cell activation.

3 - The authors often mention the suppressive effect of the resistance allele on DC function. It is not clear how the assay was designed to truly monitor suppression of T cell activation as opposed to a decrease in T cell activation. A true suppression assay must be performed to claim suppressive activity.

Minor comments:

The nomenclature for designating DC should potentially be revised in line with the recent designations. I believe that the authors are referring to a subset of cDC2, namely CD4+ cDC2. It is not clear why the authors avoid using the recent nomenclature.

References are missing to support the first statements in the introduction.

Figure 2 – There are white and black bars in the figure. It is not clear what these represent.

Why is it unexpected that the Clec4b is less expressed in the DA mice carrying the stop codon?

Figure 2b. The relative expression of Clec4b in B cells and T cells is not shown.

Figure 6. It is mentioned that the DCs suppress T cell activation. Is it truly suppression or lack of activation?

Figure 6. The data presented in panel D is very difficult to interpret. According to the labelling in the figure, the greatest effect seems to come from the genotype of the T cells. This contrasts with the information written in the text.

Panel 6E is mentioned in the text, but is not present in the figure.

The y axes in figure 7C appear to be non-linear. The interpretation of the data is difficult.

There are numerous grammatical and syntax errors throughout. In addition, the nomenclature is sometimes erroneous, not uniform, or not standard. For example, CD11bc should read CD11b/c; snps are usually abbreviated as SNPs; cells were not sorted – they were selected with magnetic beads; etc. The text and figures need to be carefully revised.

Reviewer #2: This is an interesting study where the authors describe the possible identification of a new rat autoimmune arthritis gene and suggest a mechanistic explanation. While the findings are of potential interest there are several issues in the study that were not clear, including assumptions attributed to a gene that was not clearly studied, and therefore lacked mechanistic specificity. Here are my comments including major and minor ones:

Given that the locus on chromosome 4 was identified in PIA and the congenics were protected in PIA (figure 1) it was not clear why the authors used the milder OIA to fine map the region.

More details about the breeding of the congenics should be provided as it is not clear how many generations were backcrossed and whether contaminating E3 fragments elsewhere in the genome were in fact excluded (this is critical in this study). What markers were used in the breeding?

The Clec region had been previously implicated in OIA by Lorentzen et al. In this study the authors tried to refine that association. Clec4b was implicated as the disease gene in the 200kb interval primarily because E3 and BN shared the same allele, while DA and BN had the same Clec4e allele. However, there is a co-localizing locus in a DAxF344 intercross and in congenics, and DA has the same alleles as F344 at both gene SNPs as reported in figure 1B. That raises concern about whether this is in fact the gene accounting for arthritis effect, or whether there is another gene elsewhere (E3 contamination). Additionally, it was surprising to see the authors exclude the intergenic and intronic SNPs at Clec4b, Clec4d and Clec4d simply because they were not in coding regions, particularly given that it is now known that those distant elements may regulate gene expression. Therefore, it appeared to me that there was not enough evidence to implicate only the coding SNP in Clec4b and exclude the others.

It was also surprising that the only “functional data” that really relates to Clec4b were mRNA levels. The authors should have shown changes in Clec4b protein levels (none is shown despite extensive flow cytometry looking at other proteins), or protein function, or protein localization in the plasma membrance, or in signaling pathways specifically regulated by Clec4b, or that knocking down Clec4b affected a phenotype. But none of that was shown and therefore the implication of Clec4b in the CD4+APC-regulated behavior appeared speculative. How do the authors think that Clec4b in APCs would regulate autoreactive T cell development/proliferation?

Unclear why authors use two different nomenclatures to refer to the E3 congenics: please choose one.

In addition to spleens, it would have been interesting to know whether the synovial tissue expression of the Clec genes differed between DA and congenic.

The authors should comment on whether the Clec region has been associated with rheumatoid arthritis or other autoimmune diseases in humans.

APC and DC appear to be used interchangeably in the manuscript which was a little confusing and could be changed.

Results, section 4, paragraph 2: reference figure 3c but in fact should be figure 4e.

Results, section 5, last paragraph: CD62L data not shown and it is not clear what the authors think of those results.

ConA has been shown to induce murine macrophage apoptosis: did the authors check for that? Is it possible that macrophages from the two strains differed in susceptibility to apoptosis and that was the difference in mediating T cell responses?

How is ConA affecting Clec4b expression, and how is Clec4b interfering with autoreactive T cell development and/or responsiveness?

The authors used the term ‘positional cloning’ in the title, but then refer to the SNP as ‘arthritis-associated’ in the results: please clarify.

It was not clear which strain was studied in figure 3 but it would be interesting to see a comparison between both strains.

There were also several minor text typing errors.

**Have all data underlying the figures and results presented in the manuscript been provided?**

Reviewer #1: No: The data is plotted in bar graphs showing mean and SD. It would be best if the data points were shown. Some of the representative flow cytometry profiles are missing.

Reviewer #2: No: breeding strategy and markers used were not provided.

some flow cytometry mentioned in the manuscript was also not shown.

PLOS authors have the option to publish the peer review history of their article (what does this mean?). If published, this will include your full peer review and any attached files.

Reviewer #1: No

Reviewer #2: No

---

## [Decision Letter · Decision Letter 1]

3 Feb 2020

Dear Dr Holmdahl,

Thank you very much for submitting your Research Article entitled 'Positional cloning of Clec4b - a novel regulator of bystander activation of auto-reactive T cells and autoimmune disease' to PLOS Genetics. Your manuscript was fully evaluated at the editorial level and by independent peer reviewers. The reviewers appreciated the attention to an important topic but identified some aspects of the manuscript that should be improved. 

We therefore ask you to modify the manuscript according to the review recommendations before we can consider your manuscript for acceptance. Your revisions should address the specific points made by each reviewer.

Editors comments:

The manuscript still needs to be heavily edited for English grammar, etc, and the typos in Figures fixed as noted by the reviewers.

“Positional cloning” is an inexact term that only partially describes your efforts.  You used high resolution mapping to narrow a candidate genomic region.  This was followed by sequence analysis to identify a most probable SNP in Clec4b, followed by functional studies to test that contention. 

As noted in the reviews, the evidential support that Clec4b is responsible for the phenotypes is indirect without complementation or some other direct test.  We suggest that you you temper the interpretations and conclusions in that regard. What you have accomplished is to provide a several lines of indirect support for your contention, importantly including new data with the anti-Clec4b antibody (Fig 3F).  

[LINK]

Yours sincerely,

Derry C. Roopenian

Associate Editor

PLOS Genetics

Gregory Barsh

Editor-in-Chief

PLOS Genetics

Reviewer's Responses to Questions

**Comments to the Authors:**

Reviewer #1: The authors addressed some of my concerns by providing additional data supporting the view that Clec4b contributes to autoimmunity. However, a complementation assay was not performed.

Some of the text still needs revision. This needs to be corrected by an English-speaking editor.

It is so difficult to read at times that I am not sure I understand the experimental approach.

Here are some examples (please don't limit your corrections to these few examples).

Therefor (needs an e)

"When developing tissues specificity" This is in the intro. It is not clear what is meant here.

A few sentences start with "Suggesting". This needs to be corrected.

Supernatant usually does not contain cells. Do the authors mean "flow through"?

effects vs affects

A sentence starts describing figure 6 with "Most likely" - this sentence structure needs to be revised.

Capital letters are used inappropriately in words in mid-sentence.

etc...

Reviewer #3: The manuscript entitled "Positional cloning of Clec4b - a novel regulator of bystander activation of auto-reactive T cells and autoimmune disease" proposed that Clec4b is a novel regulator expressed in CD4+ DCs and limit T cell activation. Although the authors did not clearly address the molecular mechanism mediated by Clec4b, the identification of candidate polymorphism by positional cloning is significant. To confirm the role of Clec4b directly through genetical approaches, the authors may want to perform CRISPR/Cas9-mediated complementation in DA rats and/or introduction of the same mutation in E3 rats.

**Have all data underlying the figures and results presented in the manuscript been provided?**

Reviewer #1: Yes

Reviewer #3: No: The mRNA expression of each Clec would be better to be shown as ∆∆Ct.

PLOS authors have the option to publish the peer review history of their article (what does this mean?). If published, this will include your full peer review and any attached files.

Reviewer #1: No

Reviewer #3: No

---

## [Editor Report · Decision Letter 2]

18 Mar 2020

Dear Dr Holmdahl,

Thank you very much for submitting your Research Article entitled 'Identification of Clec4b as a novel regulator of bystander activation of auto-reactive T cells and autoimmune disease' to PLOS Genetics. Your manuscript was fully evaluated at the editorial level and by independent peer reviewers. The reviewers appreciated the attention to an important topic but identified some aspects of the manuscript that should be improved.

While we continue to believe that this study deserves communication by *PLOSgenetics*, consistent with the reviewers previous comments, we believe that definitive causality of the Clec4b missense SNP continues to be overstated in the revised manuscript. 

In Abstract: “We have positionally cloned a non-sense loss of function single nucleotide polymorphism in the C-type lectin receptor, Clec4b, and have shown that it controls chronic autoimmune arthritis in rat models of rheumatoid arthritis. “

In Introduction: “We have now located the causative single nucleotide polymorphism (SNP) in the Pia7 locus where it functions in regulating the Clec4b gene.”

In Discussion: (1st para) “The identification of a single nucleotide polymorphism controlling the expression of Clec4b/Dcar in the rat has been found to be of major importance for the regulation of autoreactive T cell activation and the development of arthritis.”  

The 1^st^ paragraph of the discussion would be a good place to rationalize why the you consider the SNP to be the likely cause of the QTL. 

We ask you to modify the manuscript according to the review recommendations before we can consider your manuscript for acceptance. Your revisions should address the specific points made.

In addition, we suggest that you again have the manuscript edited for its grammar.

1) Provide a detailed list of your responses and a description of the changes you have made in the manuscript.

You can use this link to log into the system when you are ready to submit a revised version, having first consulted our Submission Checklist.

[LINK]

Yours sincerely,

Derry C. Roopenian

Associate Editor

PLOS Genetics

Gregory Barsh

Editor-in-Chief

PLOS Genetics

---

## [Editor Report · Decision Letter 3]

22 Apr 2020

Dear Dr Holmdahl,

We are pleased to inform you that your manuscript entitled "Identification  of Clec4b as a novel regulator of bystander activation of auto-reactive T cells and autoimmune disease" has been editorially accepted for publication in PLOS Genetics. Congratulations!

Yours sincerely,

Derry C. Roopenian

Associate Editor

PLOS Genetics

Gregory Barsh

Editor-in-Chief

PLOS Genetics

Comments from the reviewers (if applicable):

**Data Deposition**

http://datadryad.org/submit?journalID=pgenetics&manu=PGENETICS-D-19-01315R3

**Press Queries**

---

## [Editor Report · Acceptance letter]

28 May 2020

PGENETICS-D-19-01315R3 

Identification of *Clec4b* as a novel regulator of bystander activation of auto-reactive T cells and autoimmune disease   

Dear Dr Holmdahl, 

We are pleased to inform you that your manuscript entitled "Identification of *Clec4b* as a novel regulator of bystander activation of auto-reactive T cells and autoimmune disease  " has been formally accepted for publication in PLOS Genetics! Your manuscript is now with our production department and you will be notified of the publication date in due course.

With kind regards,

Matt Lyles

PLOS Genetics

On behalf of:
